# Rubella Seroprevalence Boost in the Pediatric and Adolescent Population of Florence (Italy) as a Preventive Strategy for Congenital Rubella Syndrome (CRS)

**DOI:** 10.3390/vaccines8040599

**Published:** 2020-10-12

**Authors:** Beatrice Zanella, Sara Boccalini, Benedetta Bonito, Marco Del Riccio, Federico Manzi, Emilia Tiscione, Paolo Bonanni, Angela Bechini

**Affiliations:** 1Department of Health Sciences, University of Florence, 50134 Florence, Italy; beatrice.zanella@unifi.it (B.Z.); sara.boccalini@unifi.it (S.B.); benedetta.bonito@unifi.it (B.B); emilia.tiscione@unifi.it (E.T.); paolo.bonanni@unifi.it (P.B.); 2Medical Specialization School of Hygiene and Preventive Medicine, University of Florence, 50134 Florence, Italy; marco.delriccio@unifi.it (M.D.R.); federico.manzi@unifi.it (F.M.); alessandra.ninci@unifi.it (W.D.G.); 3Meyer Children’s Hospital, 50139 Florence, Italy; francesco.puggelli@meyer.it; 4AUSL Toscana Centro, 50122 Florence, Italy; giovanna.mereu@uslcentro.toscana.it

**Keywords:** rubella, CRS, vaccination coverage, pediatric, adolescent, seroprevalence, Italy, Florence, MMRV vaccination, elimination

## Abstract

Background: Despite the availability of an effective vaccine since the 1970s, rubella disease and, importantly, congenital rubella syndrome (CRS) remain a public health concern. The aim of this study was to analyze the rubella seroprevalence in the children population of the province of Florence and compare the obtained results to a previous survey conducted in 2005–2006. Methods: A qualitative measurement of anti-rubella antibodies was performed on 165 sera using the enzyme-linked immunosorbent Assay (ELISA). The anamnestic and vaccination status was also collected. Results: Our study highlighted a very high rubella seroprevalence (85–100%) in our enrolled population. In the vaccinated group (153/165), 98.7% of them were positive to rubella antibodies. Conclusions: Our study showed the highest seroprevalence rate reached in the province of Florence for rubella in the last 15 years, thanks to the several successful vaccination campaigns promoted in the Tuscany region. Our findings in pediatric and adolescent subjects are a key factor in preventing CRS in adult life, specifically in childbearing women. Thus, the set goal will be to keep the awareness about the vaccination for this preventable disease high.

## 1. Introduction

Rubella was first described in the 18th century in Germany (hence known also as German measles) as a mild exhantematic disease predominant in children and young adults [1]. However, towards the mid 19th century, some newborns’ deficits started to be associated with maternal rubella; this was firstly reported in Australia [2], followed by other worldwide studies which described congenital cataracts, heart disease and deafness linked to rubella in infants [3,4,5]. Thus, the typical triad of congenital rubella syndrome (CRS) was established (Gregg’s triad) [2]. The following rubella outbreak in Europe and the United States of America added other symptoms to the known ones, including: hepatitis, splenomegaly, thrombocytopenia, encephalitis, mental retardation and several other anomalies [6]. The urgent need to find a rubella vaccine was met in 1970 when the live attenuated vaccine became available in Europe and North America [7,8,9]. Rubella virus (RV) is the only member of the *Rubivirus* genus within the Togaviridae family and was first observed in 1967 revealing a pleomorphic nucleocapsid core enveloped with a host-derived lipid membrane. The average rubella incubation time is 14 days (range is 12–21 days), and during the first week after virus exposure, symptoms are absent. Additionally, during the second week, there may be a prodromal infection characterized by low-grade fever (<39.0 °C), malaise, mild coryza and mild conjunctivitis, which is more common in adults. Children usually develop few or no constitutional symptoms. At the end of the incubation period, a maculopapular erythematous rash appears on the face and neck. The rubella rash occurs in 50–80% of rubella-infected subjects, and it is sometimes misdiagnosed as measles or scarlet fever. [10,11]. Rubella disease is generally mild, and a few complications may occur, including arthralgia and chronic arthritis (common) [12], and thrombocytopenia and encephalitis (less common) [13,14]. Importantly, the real threat arises when RV infects the fetus through the maternal placenta; particularly critical is the first trimester when infection can lead to miscarriage or CRS; the same happens if the infection occurs just before conception [15]. The risk of CRS is related to the gestational age at the time of maternal infection. Specifically, the first trimester is the most dangerous period for the fetus to develop abnormalities (i.e., sensorineural deafness, cataracts, pigmentary retinopathy and patent ductus arteriosus); then, the incidence of fetal disease declines during the next month, and, between the 16th and the 20th weeks, only deafness has been reported as a complication [11]. Despite the availability of a safe, effective and inexpensive vaccine, rubella infections remain among the leading causes of preventable congenital birth defects globally. Three WHO Regions (Americas, Europe and Western Pacific) have set goals to eliminate rubella and CRS. As of January 2019, the number of countries in the world which have definitely eliminated rubella was 76, while 118 are still considered to be endemic. Moreover, in the European Region, 11 countries are still endemic for rubella and 9 for both measles and rubella diseases, the latter including Poland, Romania, Serbia, France, Germany and Italy [16]. The key prevention tool remains vaccination: WHO has recommended maintaining 95% coverage for measles–mumps–rubella (MMR) vaccination in children in adolescents and women of childbearing age [17]. Specifically, the vaccine has proven to be highly effective: >95% of subjects vaccinated with a single dose are protected against both clinical rubella and viremia for at least 15 years [18]. Currently, a live attenuated vaccine in two formulations is available in Italy: trivalent measles–mumps–rubella (MMR) and quadrivalent measles–mumps–rubella–varicella MMRV. The Italian vaccination schedule recommends the administration of one dose between the 13rd and the 15th month, and the second at 5–6 years of age. Children who fail to respond to the first dose (especially for the measles component) will probably respond to the second one, thus ensuring a seroprevalence rate ≥ 95% after two doses if the first dose is given at nine months, or ≥99% if the first dose is given at 12 months or older [19]. WHO also recommends improving the surveillance by rigorous case investigation and laboratory confirmation of suspected rubella cases [17]. Looking at the Italian scenario, the first National Plan for the Elimination of Measles and Congenital Rubella (NPEMcR) of 2003–2007 had an incidence target for congenital rubella set at <1/100.000 live births. Nevertheless, this goal was not achieved in 2010, so the plan was been implemented in 2011. In the new NPEMcR 2010–2015, both measles and rubella surveillance has been strengthened to effectively monitor the progress towards the elimination of measles and the prevention of CRS: the goal for vaccination coverage (VC) after two vaccine doses was set at 95%, and the threshold of susceptible childbearing women was set at <5% [20]. Laboratory tests of all suspected cases, with particular attention to rubella cases in pregnant women, started in 2005, when CRS became a mandatory notifiable disease, and a catch-up campaign to vaccinate school-aged children (7–14 years old) was conducted in all Italian regions [19,20,21]. According to the latest report, 12 rubella cases have been notified in 6 Italian regions (in the period January–June 2020), and the median age was 33 years, varying in a range of 1–59 years [22]. Importantly, in Tuscany, the rubella incidence has dramatically dropped in the last 20 years: in 2018, only three rubella cases have been reported, and two of them were not vaccinated. Moreover, in Tuscany, the highest rubella VC at 24 months (95.3%) was reached in 2019 [23]. 

This study was part of a wider project which implied the analysis of the seroprevalence of vaccine-preventable infectious diseases including Measles [24], Varicella, Hepatitis A and B [25] and Tetanus. The aim of this study was to investigate the rubella seroprevalence in pediatric and adolescent (1–18 years) subjects in the province of Florence, in order to discover possible pockets of susceptibility, especially in female adolescents. Moreover, we aimed to compare the obtained data with those of a previous seroepidemiological survey (2005–2006) within the same geographical area in order to describe the temporal trend of the immunity level against rubella for the pediatric and adolescent population [26]. Furthermore, we relate the vaccination status of each subject to the results of seroprevalence assays, and assess if it was due to a natural infection or due to the vaccination, and assess the VC with the administration of one or two vaccine doses. 

## 2. Materials and Methods 

This rubella seroprevalence analysis is part of a wider seroepidemiological project carried out by the Department of Health Sciences of the University of Florence started in 2017. The study was conducted in accordance with the Declaration of Helsinki and the protocol was approved by local ethics committees (Project identification code: DSS-UNIFI, n. registro pareri 98/2017). In a post-hoc analysis, the calculated sample size was 164 sera estimating the expected seroprevalence of anti-rubella virus equal to 96%, with an accuracy of 3% and a confidence level of 95%. The number of sera approximately represents 0.1% of the resident population of Florence aged 1–18 years of a total of 166,644 subjects in 2017 in the same age group [27]. This is also proportionally related to the population composition for each age group and sex. Thus, no further standardization was required.

Prior to sera collection, we excluded subjects who were non-resident in the province of Florence; immunocompromised patients or under immunosuppressive treatment; those with an acute infectious disease among measles, rubella, varicella, hepatitis A and hepatitis B in the previous two weeks; and those who had received a blood transfusion within the six months before the study.

### 2.1. Enrollment and Conservation of Sera Sample

The enrollment of the study population and the collection of the sera took place at the blood sampling center of Meyer’s Children Hospital, starting in December 2017 and finishing in April 2018. Prior to the enrollment, enrolled subjects’ parents or guardians gave written consent for the inclusion in the study.

All the collected sera samples were centrifuged (1600 rpm at 4 °C), and the recovered sera were stored at −20 °C until tested for rubella IgG.

### 2.2. Confirmation of Anamnestic and Vaccination Status

The National Registry of Notifications for Infectious Diseases (SIMI, software: Epi Info, Rome, Italy) was consulted to collect the anamnestic status of each subject.

Through the vaccination registers SISPC (Collective Prevention Healthcare Information System) (Consortium Metis, Tuscany, Italy) and Caribel (Aster, Tuscany, Italy), respectively, the current and the previous VC software used in the Tuscany Region, we were able to collect the vaccination status for rubella for each subject. In particular, we retrieved the number of rubella vaccine doses, the year of the last dose and, if available, the type of the last administered vaccine. 

### 2.3. Serological Analysis 

To perform a qualitative measurement of anti-rubella antibodies, we used the commercial Enzygnost^®^ Rubella anti-virus/IgG enzyme-linked immunosorbent Assay (ELISA). All the collected sera were tested for anti-rubella IgG antibodies and classified according to the cut-off values below: Anti-rubella/IgG negative ΔA < 0.100 (cut-off);Anti-rubella/IgG positive ΔA > 0.200;Anti-rubella/IgG equivocal 0.100 ≤ ΔA ≤ 0.200.

The ΔA value was calculated as the difference of the absorbance obtained for each sample and the absorbance value of the same sample containing the control rubella antigen. 

If the result was positive, the sample was considered positive: on the other hand, if the result was negative, the sample was considered negative. Finally, if the sample gave an equivocal outcome, it was then analyzed a second time. 

### 2.4. Statistical Analysis

The serological IgG rubella results were collected into an Excel database and assessed through a descriptive analysis for antibody presence (as positive and negative), sex, nationality, age group, vaccination status, number of vaccine doses received and time elapsed since the last dose of vaccine received. Rubella seroprevalence rates were calculated along with their corresponding 95% Confidence Interval (CI).

We classified as Italians all the subjects with Italian nationality and all the subjects with a foreign nationality either born in Italy or adopted. Subjects born abroad with a double nationality (Italian–foreign) or a foreign nationality were classified as Not-Italian. 

Fisher’s exact test and Cochrane–Armitage test for trend was used to compare groups. The statistical analyses were conducted using RStudio 1.2.5033 (RStudio Team, 2019. RStudio: Integrated Development for R. RStudio, Inc., Boston, MA, USA, URL http://www.rstudio.com/). A *p* < 0.05 was considered statistically significant.

## 3. Results

The number of the enrolled subjects broken down by age groups is reported as follows: 40 in the age group 1–4 years old; 48 in the age group 5–9 years old; 50 in the age group 10–14 years old; and 27 in the age group 15–18 years old. The participants resided in 35 different districts of Florence, and about 48% of them were living in the City of Florence. The percentages of males and females in the group were 53.3% and 46.7%, respectively. The vast majority of the enrolled subjects were Italians (90.3%), and the remaining part were non-Italians (9.7%). 

### 3.1. Rubella Seroprevalence Analysis 

Table 1 shows the percentage of the positive and negative subjects in relation to their sex (male/female) and their nationality (Italian/Not-Italian). There was no equivocal sample. Amongst the 165 samples, 158 were positive (95.8%; 95% CI: 92.7–98.9) and 7 were negative (4.2%; 95% CI: 1.2–7.3). No significant difference was found between male and female and between Italian/non-Italian subjects (*p* = 0.708 and *p* = 0.512 respectively). 

The distribution of positive and negative subjects in the different age groups is shown in Figure 1. In the first group (1–4 years), 34/40 subjects were positive (85%; 95% CI: 73.9–96) and 6/40 were negative (15%; 95% CI: 3.9–26.1); in the second group (5–9 years), 47/48 were positive (97.9%; 95% CI: 93.9–100) and 1/48 was negative (2.1%; 95% CI: 0–6.1); finally, in the last age groups (10–14 and 15–18), all the subjects were positive: 50/50 and 27/27, respectively. 

### 3.2. Rubella Notification, Vaccination Status and Seroprevalence Assessment

According to the SIMI, none of the enrolled subject was notified for rubella disease. The majority of them were vaccinated (153/165 corresponding to 92.7%), whilst the unvaccinated were 12/165 corresponding to 7.3%. Table 2 summarizes these results by age group and seroprevalence. Overall, in the vaccinated group, 151/153 subjects were positive (98.7%), and 2/153 were negative (1.3%). The subjects vaccinated were all positive except 2 negative ones in the age group 1–4 years old. On the other hand, in the unvaccinated group, 7/12 subjects were positive (58.3%), and 5/12 were negative (41.7%). Unvaccinated subjects were all negative in the age group 1–4 years old and 5–9 years old, and were all positive in the age groups 10–14 and 15–18 years old. 

### 3.3. Seroprevalence Assessment by Number of Vaccine Doses and Time Elapsed since Last Vaccination


We then retrieved the number of vaccine doses administered to each subject, which resulted in 52/153 vaccinated who had received one dose of vaccine, and, amongst them, 50 were positive (96.2%) and 2 were negative (3.8%). On the other hand, 101/153 vaccinated had received two/three doses and all of them were positive (100%). The Cochrane–Armitage test for trends shows association (*p* < 0.001) between the result (negative/positive) and the number of doses (0/1/2). 

The rubella seroprevalence was consistent and high (100%) over time, up to 14 years since the last vaccination received (Figure 2). The only two negative subjects had received the vaccine dose either in the same year of the sera collection or in the previous one. 

## 4. Discussion

In this study, we assessed the rubella seroprevalence in a pediatric and adolescent population resident in the province of Florence. Specifically, the aim of this study was not only to assess whether the vaccination promotion campaigns were being successful but also to use it as a potential preventive tool for CRS. Indeed, a major public health concern is about women who are not aware of their immunization status against rubella (38%), with a noticeable difference within Italian Regions. The vast majority of anti-rubella vaccinated women are in the age group 18–24 (59%) years old (thanks to the NPEMcR), whilst the percentage decreases in 25–34-year-old (47%) and 35–49-year-old women (36%) [28].

The distribution of our sample population is a good representation of the general Italian scenario since almost 17% are Not-Italian citizens, and in Italy, children born from at least one foreign parent in 2018 were ~20% [29]. No notification of rubella disease was reported in any subject of our study; this result was expected since the average incidence age of rubella in Italy was 33 years old [22]. Comparing the data obtained in this study with the one conducted previously [26], the percentage of susceptible (negative) subjects is remarkably lower in the current survey: 15% in the age group 1–4 years vs. 21.2% in 2005–2006; 2.1% in the age group 5–9 years old vs. 11.7% in 2005–2006. Moreover, no susceptible subject was found in the 10–14 nor in the 15–18-year-old groups, whereas 13% and 15.9% of seronegative samples were, respectively, found in 2005–2006. Thus, the WHO guidelines to maintain the susceptible subject threshold < 5% in females of childbearing age is reached since adolescence [30]. 

The number of negative subjects in this age group was 6 out of 40: two of them were vaccinated, while four of them were not. The two vaccinated yet negative subjects are both Italian one-year-old children who probably had still not developed a measurable level of antibodies: rubella IgG are typically detectable after 8 weeks from vaccine administration; otherwise, these subjects may be not-responder cases (generally 5%). 

Overall, the unvaccinated are 7.3% of our enrolled population, and they are mostly in the 10–14 years old group (50% of the complete unvaccinated group) and the 1–4 years old (33.3%) group. However, not all of them are negative to serological test. The unvaccinated and negative subjects are all in the first two age groups (1–4 and 5–9 years old). Missing vaccination is involved in the global phenomenon of the parental vaccine hesitancy (“no-vax movements”) in which both religious and moral beliefs, complacency and skepticism created the perception of vaccination as a scary and unnecessary practice [31]. In order to fight and contrast the consequential decreasing trend in immunization coverage in 2017, an Italian national law, the “Vaccine Decree” [32], introduced MMR vaccination as mandatory in schools for subjects between zero and 16 years of age. Moreover, among the unvaccinated yet positive to serological test subjects, there are 7 subjects: 6 are in the 10–14 age group (Italian, Philippine, Senegalese, Moroccan and Romanian nationalities) and one is in the 15–18 age group (Italian nationality). In all the above countries, despite VCs being quite high (85–90%) [33,34,35,36], except for the Philippines (~70%) [37], rubella cases have been reported in nearly every country. Thus, it is likely that the Philippine child had not received the vaccination dose in the Philippines, whereas the other children probably had an undiagnosed or misreported disease, or the vaccination was not reported. Indeed, a recent study highlighted the fact that more than half of internationally adopted children referred to Meyer Children’s University Hospital were unprotected against MMR-V [38]. Since rubella usually presents mild and not disease-specific symptoms, it is difficult to diagnose, and even more challenging is CRS diagnosis; thus, a retrospective case-finding study estimated the degree of CRS underreporting as 53% for the period 2010–2014 [39]. 

Concerning the number of doses, even in subjects who received only the first vaccine dose, the seroprevalence value reached was already high (96.2%), and progressively, it reached the total (100%) in subjects who had received two or more doses. As widely discussed in the literature and confirmed by our statistical analysis, the administration of a second dose may increase the probability to seroconvert anti-rubella IgG in non-responder subjects [10,11,40]. Furthermore, catching-up the unvaccinated subjects and/or those who received only one dose is a crucial aspect to increase the global immunity level; the most represented and yet hardest reachable category in our study is the adolescents. A few papers suggest the direct involvement of teenagers in vaccination by using social networks and text messages as well as sending e-mails to their parents; these communication strategies have been found to be promising [41,42]. 

## 5. Conclusions

Our study highlighted the highest seropositivity against rubella in a 1–18-year-old sample population in the province of Florence in the last 15 years. This result gives not only excellent feedback of the numerous vaccination campaigns promoted but also strengthens the herd immunity, allowing the protection of immunocompromised subjects who cannot be vaccinated. Moreover, the high anti-rubella seroprevalence found in pediatric and adolescent subjects is an important factor to prevent CRS in adult life, specifically in childbearing women. Although rubella is a mandatory notified disease and many efforts have been made to vaccination catching-up, rubella and CRS are still not eliminated. Therefore, it is important to follow the Institutional guidelines in order to maintain a high level of rubella monitoring and surveillance as well as maintain the VC above the recommended threshold (>95%) (as occurred in Tuscany in recent years) [43]. 

## Figures and Tables

**Figure 1 vaccines-08-00599-f001:**
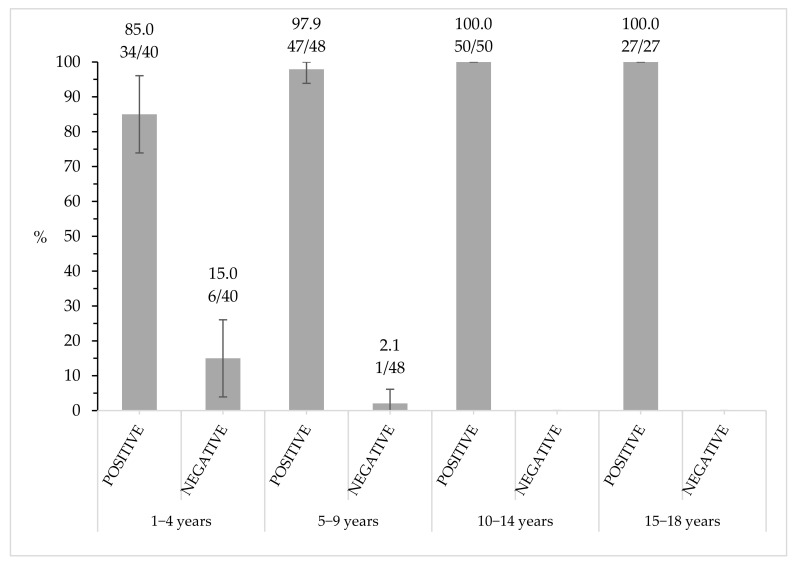
Percentage of rubella immunity distribution in the age groups.

**Figure 2 vaccines-08-00599-f002:**
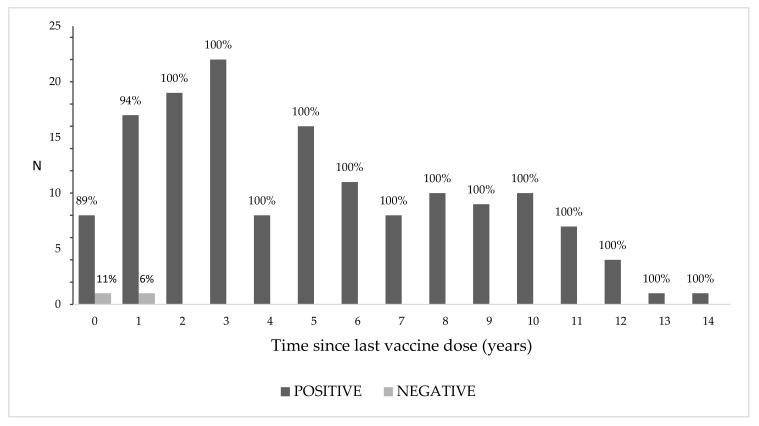
Rubella seroprevalence by time since last dose (years). Percentages are shown above each bar.

**Table 1 vaccines-08-00599-t001:** Anti-rubella seroprevalence in the enrolled population. (n/N: number of subjects/total).

Anti-Rubella Seroprevalence		
Group	Positive % (n/N)	Negative % (n/N)
Overall	95.8 (158/165)	4.2 (7/165)
Male	96.6 (84/87)	3.4 (3/87)
Female	94.9 (74/78)	5.1 (4/78)
Italian	96.0 (143/149)	4.0 (6/149)
Not-Italian	93.8 (15/16)	6.2 (1/16)

**Table 2 vaccines-08-00599-t002:** Vaccination status and seroprevalence assessment of the enrolled subjects.

Vaccination Status	Age Group (Years)	Positive % (n/N)	Negative % (n/N)	Total % (n/N)
Vaccinated		98.7 (151/153)	1.3 (2/153)	92.7 (153/165)
	1–4	94.4 (34/36)	5.6 (2/36)	23.5 (36/153)
	5–9	100.0 (47/47)	0.0 (0/47)	30.7 (47/153)
	10–14	100.0 (44/44)	0.0 (0/44)	28.8 (44/153)
	15–18	100.0 (26/26)	0.0 (0/26)	17.0 (26/153)
Unvaccinated		58.3 (7/12)	41.7 (5/12)	7.3 (12/165)
	1–4	0.0 (0/4)	100.0 (4/4)	33.3 (4/12)
	5–9	0.0 (0/1)	100.0 (1/1)	8.3 (1/12)
	10–14	100.0 (6/6)	0.0 (0/6)	50.0 (6/12)
	15–18	100.0 (1/1)	0.0 (0/1)	8.3 (1/12)

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
