# Peer review of "Rubella Seroprevalence Boost in the Pediatric and Adolescent Population of Florence (Italy) as a Preventive Strategy for Congenital Rubella Syndrome (CRS)"

_vaccines, 2020, doi:10.3390/vaccines8040599_

Round 1

Reviewer 1 Report

The manuscript by Zanella et al., reports the seroprevalence of Rubella in a small population of 165 individuals in the age group of 1-18 years, with reporting correlation of positive and negative results with vaccination status. 

The study provides an insight into the status of immunity against Rubella that is important for the evaluation of the success of vaccination efforts.

Specific comments: 

The title is over concluding the findings of the study since this is a single point evaluation of the antibody status and not over a long period of time and with no reference/attempts to the evaluation of the congenital problems due to Rubella infection. 

 The title needs to be corrected something like "Rubella seroprevalence in the pediatric and adolescent population of Florescence (Italy)- a cross-sectional study"

Author Response

Response to Reviewer 1 Comments
The manuscript by Zanella et al., reports the seroprevalence of Rubella in a small population of 165 individuals in the age group of 1-18 years, with reporting correlation of positive and negative results with vaccination status. The study provides an insight into the status of immunity against Rubella that is important for the evaluation of the success of vaccination efforts.
Point 1: The title is over concluding the findings of the study since this is a single point evaluation of the antibody status and not over a long period of time and with no reference/attempts to the evaluation of the congenital problems due to Rubella infection.
The title needs to be corrected something like "Rubella seroprevalence in the pediatric and adolescent population of Florescence (Italy) - a cross-sectional study”
Response 1: We think the title we proposed already includes the information about the “cross-sectional study” that you suggest. Moreover, your fellow reviewer particularly likes our title, claiming it displays the main objective of the study. We would then prefer to keep the original title.

Reviewer 2 Report

Dear Authors

The study of Zanella et al. reports rubella seroprevalence in the children population of the province of Florence. The overall manuscript presentation was impressive and interesting. The results are important in preventing congenital rubella syndrome in adult life, specifically in childbearing women.

Title is clear and informative; it displays the main objective of the study.

The abstract is sectioned. It contains focused background with clear objective.

Overall, the methodology applied seems appropriate for the purpose and it is already well-established in these types of studies.

The interpretation of the results is correct. It will be interesting include 95% confidence intervals.

The reference list cover the relevant literature adequately and in an unbiased manner.

Author Response

Response to Reviewer 2 Comments
Dear Authors
The study of Zanella et al. reports rubella seroprevalence in the children population of the province of Florence. The overall manuscript presentation was impressive and interesting. The results are important in preventing congenital rubella syndrome in adult life, specifically in childbearing women.
Title is clear and informative; it displays the main objective of the study.
The abstract is sectioned. It contains focused background with clear objective.
Overall, the methodology applied seems appropriate for the purpose and it is already well-established in these types of studies.
Point 1: The interpretation of the results is correct. It will be interesting include 95% confidence intervals.
Response 1: Thank you for your comment. We added a section about the CI calculation in “Materials and Methods” (lines: 159-160). We also calculated the 95% CI for the overall seroprevalence in the text about Table 1 (lines 178-179) and Figure 1 (lines: 184-186).
The reference list cover the relevant literature adequately and in an unbiased manner.

Reviewer 3 Report

Dear Authors,

I appreciate your work; however I have few concerns which need to be addressed;

1. I would see the vaccinated group consist of 153 subjects however the unvaccinated group is very limited, subjects of 12 only included making it statistically insignificant to correlate the data. So, increase the population size for unvaccinated group over 100 subjects.
2. Apart from that testing only anti-rubella antibodies (IgG), other parameters would be tested. Also isototyping will add more significance to the work.
3. a clinical data table should be included, apart from the table1 which does not have detailed information.

Author Response

Response to Reviewer 3 Comments
Dear Authors,
I appreciate your work; however I have few concerns which need to be addressed
Point 1: I would see the vaccinated group consist of 153 subjects however the unvaccinated group is very limited, subjects of 12 only included making it statistically insignificant to correlate the data. So, increase the population size for unvaccinated group over 100 subjects.
Response 1: Thank for this comment, this allow us to clarify our study population. The information about the immunization status of each enrolled subject was retrospectively collected consulting the vaccination register. We also investigated the notification recorded into the infectious diseases surveillance system for rubella disease. This let us to correlate the serological results with the vaccination or the anamnestic status. We enrolled healthy subjects who met the inclusion and exclusion criteria, but the immunization status was not considered as one of the inclusion/exclusion criteria: we were not interested in the enrollment of two cohorts of individuals, such as vaccinated and unvaccinated.
Point 2: Apart from that testing only anti-rubella antibodies (IgG), other parameters would be tested. Also isototyping will add more significance to the work.
Response 2: Thanks for the comment. The manuscript reported the results obtained by an observational – cross sectional study, in particular a seroprevalence study. For this purpose, the methodological approach most widely used is testing the presence of anti-rubella IgG antibodies in a sample of sera collected from the general population. As we detailed in “Materials and Methods”, we excluded subjects who had the acute infectious disease in the previous two weeks (including rubella), so the serological analysis were carried out not for a diagnostic purpose. For this reason, the assessment of anti-rubella IgG antibodies is adequate for the aim of the study.
Point 3: clinical data table should be included, apart from the table 1 which does not have detailed information.
Response 3: As we previously underlined in Response 1, all the subjects included in the study were healthy at the time of the enrollment. Children were enrolled at the blood sampling center of Meyer’s Children University Hospital where they went for routine blood testing and they were not hospitalized. Thus, we did not collect any clinical data.
